# Graphene controlled Brewster angle device for ultra broadband terahertz modulation

Zefeng Chen[1], Xuequan Chen[1], Li Tao [1], Kun Chen[1], Mingzhu Long[1], Xudong Liu[1], Keyou Yan [1], Rayko I. Stantchev[1], Emma Pickwell-MacPherson[1,2] & Jian-Bin Xu [1]

Terahertz modulators with high tunability of both intensity and phase are essential for effective control of electromagnetic properties. Due to the underlying physics behind existing approaches there is still a lack of broadband devices able to achieve deep modulation. Here, we demonstrate the effect of tunable Brewster angle controlled by graphene, and develop a highly-tunable solid-state graphene/quartz modulator based on this mechanism. The Brewster angle of the device can be tuned by varying the conductivity of the graphene through an electrical gate. In this way, we achieve near perfect intensity modulation with spectrally flat modulation depth of 99.3 to 99.9 percent and phase tunability of up to 140 degree in the frequency range from 0.5 to 1.6 THz. Different from using electromagnetic resonance effects (for example, metamaterials), this principle ensures that our device can operate in ultra-broadband. Thus it is an effective principle for terahertz modulation.

[1] Department of Electronic Engineering, The Chinese University of Hong Kong, Hongkong 999077, China. [2] Physics Department, Warwick University, Coventry CV4 7AL, UK. These authors contributed equally: Zefeng Chen, Xuequan Chen. Correspondence and requests for materials should be addressed to E.P.-M. (email: e.pickwell.97@cantab.net) or to J.-B.X. (email: jbxu@ee.cuhk.edu.hk)

Terahertz technology shows promise for many applications including imaging, spectroscopy, and communications[1–3]. These applications drive the need for THz modulators with high modulation depth, broad operation bandwidth, and high modulation speed[4,5]. Manipulating THz light to achieve both high modulation depth and broadband response in a device is a great challenge. One major limitation of the state-of-the-art THz modulator is the narrow bandwidth; current technologies rely primarily on narrow band resonance effects[6–13] (such as metamaterials, plasmonic material, and so on) for enhancing the interaction between the THz wave and the tunable conductivity materials. Semiconductor hybrid structures are developed, but these architectures require the assistance of high-power optical pumping. Another issue is the low modulation speed of the non-solid-state devices, such as the modulators based on ion-gel gate or liquid crystal.

Graphene is an ideal material for photoelectronic devices due to its unique properties[14–19]. It has attracted great attention in the THz region because of its tunable conductivity, typically from comparatively high-resistance state to semi-metal state[20–23]. The first graphene THz modulator is demonstrated by Sensale-Rodriguez et al.[24] with an ultra-broad operation band and a modulation depth of 15% using monolayer graphene. To improve the modulation depth, complex optical resonance structures (e.g., cavity, metasurface, and metamaterials)[25–27], semiconductor hybridization[28,29], and ion gel[30,31], are assembled to work together. However, these structures could not overcome the above-mentioned problems. Therefore, there is still a lack of an effective fundamental approach to achieving a broadband device with deep and wideband modulation range.

Here, as a proof of concept, we demonstrate a solid-state modulator with near-perfect tunability, ultra-broad operation bandwidth, and fast modulation speed based on tunable Brewster reflection at graphene/quartz surface. It is well known that $p$-polarization light experiences zero reflection when it is incident to a medium at the Brewster angle and a $\pi$-phase shift occurs to the reflected light when the incident angle jumps across the Brewster angle. Furthermore, the Brewster angle can be tuned by varying the surface conductivity of the medium[32,33]. Based on this fundamental principle, we experimentally employ an atomic layer of graphene to tune the surface conductivity and demonstrate that the Brewster angle of a graphene–quartz device can be tuned from 65 to 71° by varying the conductivity of graphene. By properly selecting the incident angle, the device can work as a THz-intensity modulator with spectrally flat modulation depth of 99.3–99.9% in a wide range from 0.5 to 1.6 THz, and operate as a phase modulator with tunability higher than 140°. Compared to the high-performance THz modulators based on light-trapping effects, this principle ensures that the device can operate as an ultra-broadband spectrally flat terahertz modulator. Actually, the reported bandwidth in this work is mainly limited by the effective bandwidth of the THz system. An even broader bandwidth can be theoretically extrapolated from the operation principle. Our device presents a THz modulation based on a graphene-controlled Brewster angle device and achieves record performance at room temperature.

## Results

**Physical model**. The physical model is demonstrated in Fig. 1. Figure 1a shows the operation principle of the device. Graphene with conductivity $\sigma_g$ is placed on the dielectric silica substrate. Light with $p$-polarization is incident from air onto the graphene/silica medium. According to Maxwell's equations, together with the boundary conditions (Supplementary Note 1 and Supplementary Figure 1), the reflection coefficient for $p$-polarization light can be expressed by Eq. (1).

$$r_p = \frac{\sqrt{\varepsilon_s \mu_s}\cos\theta_i - \sqrt{\varepsilon_0 \mu_0}\cos\Phi + Z_0 \sigma_g \cos\theta_i \cos\Phi}{\sqrt{\varepsilon_s \mu_s}\cos\theta_i + \sqrt{\varepsilon_0 \mu_0}\cos\Phi + Z_0 \sigma_g \cos\theta_i \cos\Phi}, \quad (1)$$

$$\sigma_g = i\frac{e^2 E_F}{\pi\hbar}\frac{i}{\omega + i\tau^{-1}}, \quad (2)$$

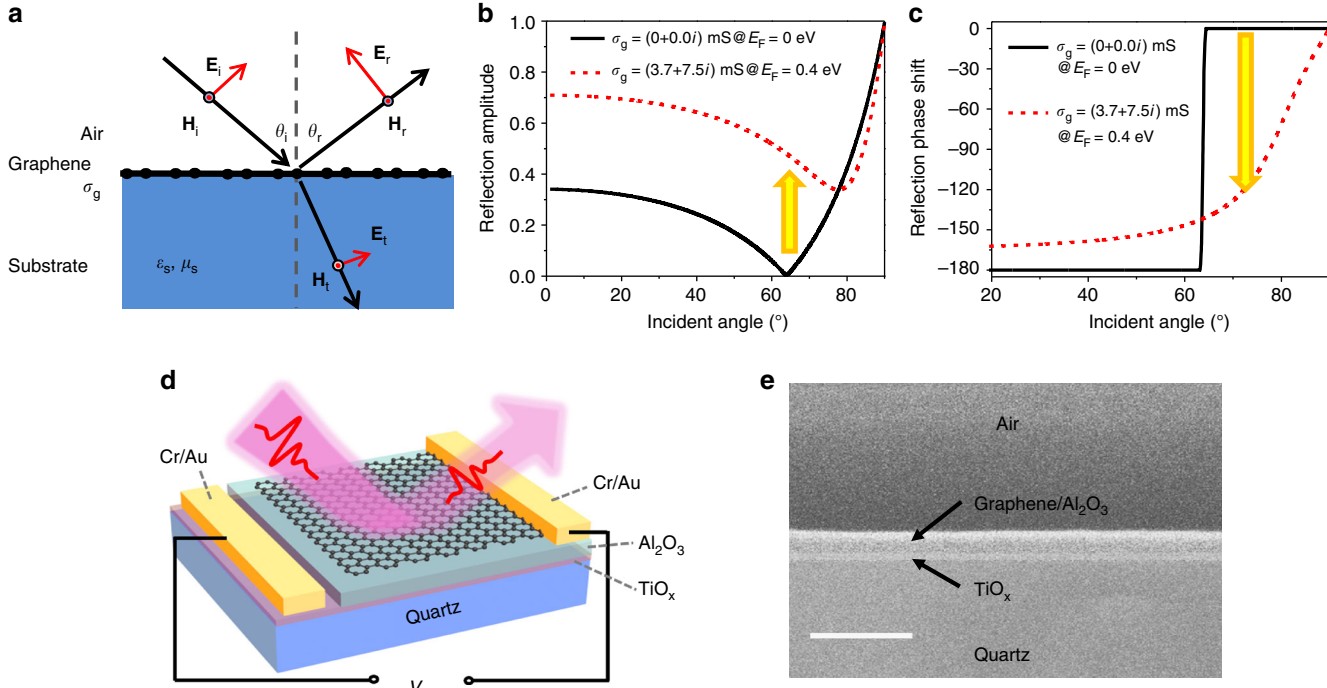

**Fig. 1** Optical arrangement and device configuration. **a** Optical path diagram of the incident light from air to graphene/substrate medium. **b** The reflection amplitude and (**c**) phase as a function of incident angle, when the graphene conductivity is at 0 and 3.7 mS. The yellow arrows indicate the modulation of intensity and phase. **d** The three-dimensional (3D) diagram of the device, and (**e**) the SEM image of the cross section of the device. The scale bar is 5 μm

where $\theta_i$ and $\Phi$ are the incident angle and refraction angle, respectively; $Z_0$ is the impedance of air; $\varepsilon_s$, $\mu_s$ are the permittivity and permeability of the silica substrate and $\varepsilon_0$, $\mu_0$ are the permittivity and permeability of air. We used the Drude model to describe the graphene conductivity, which can be expressed by Eq. (2)[34,35]. $\tau^{-1} = \frac{ev_F^2}{E_F\mu_c}$ is the damping rate of carriers. In this model, we use the ideal value of 10,000 cm$^2$ V$^{-1}$ s$^{-1}$ as the carrier mobility $\mu_c$ of graphene[36,37], and assume that the Fermi level of the graphene, $E_F$, ranges from 0.00 to 0.35 eV. $v_f$ is the Fermi velocity and $\hbar$ is the reduced Planck constant. The parameters of the substrate are $\varepsilon_s = 4$ and $\mu_s = 1$. The modulation depth in this model is defined as

$$\mathrm{MD} = \left(1 - \frac{|r|^2}{|r_{\max}|^2}\right) \times 100\%, \qquad (3)$$

where $r$ is the reflection coefficient, and $r_{\max}$ is the baseline defined as the maximum reflection value in this active system. As an example here, we show the result at 0.8 THz.

We first present the simulation results based on the physical model in Fig. 1a. When the Fermi level of graphene is at the Dirac point, the conductivity of the graphene becomes the minimum. In this case, Eq. (1) returns to the traditional Fresnel equation for reflection. At the Brewster angle, the magnitude of reflection intensity reaches zero, as shown in Fig. 1b. In addition, the phase shift of reflection light is switched from $\pi$ to 0, when the incident angle changes from $\theta_i > \theta_B$ to $\theta_i < \theta_B$, as shown in Fig. 1c. While in the case of non-zero Fermi level, the Brewster angle (defined as the incident angle that gives the smallest reflection amplitude in this case) is shifted and determined by $\sqrt{\varepsilon_s\mu_s}\cos\theta_i - \sqrt{\varepsilon_0\mu_0}\cos\Phi - Z_0\sigma_g\cos\theta_i\cos\Phi = 0$, if ignoring the imaginary part of graphene's conductivity. The term introduced by the conductivity, $Z_0\sigma_g\cos\theta_i\cos\Phi$, allows the Brewster angle to be tunable. Because the conductivity of graphene is a complex number, the reflection amplitude at the Brewster angle is not zero (Fig. 1b) and the phase changes gradually with the incident angle close to the Brewster angle (Fig. 1c), which is a little different from the abrupt change for conventional Brewster angle reflection. As shown in Fig. 1, when the conductivity changes from 0 to 3.7 mS, the Brewster angle shifts from 63 to 79°. When the incident angle at 63° is fixed, the reflection amplitude increases from 0 to ~0.52, as indicated by the yellow arrow in Fig. 1b. In this case, the modulation depth is up to 100% due to the perfect "off-state" ($|r|=0$, at 63°). The insertion loss of this device, $\lg_{10}(|r_{\max}|^2)$, is −5.6 dB. The insertion loss can be reduced to −3 dB through using higher-quality graphene or composing two-monolayer graphene (Supplementary Figure 2 and Supplementary Note 2). When the incident angle is fixed at 68°, a phase modulation range, as large as about 140°, can also be achieved, as indicated by the yellow arrow in Fig. 1c.

The real device in this work is schematically shown in Fig. 1d, e. A graphene/Al$_2$O$_3$/TiO$_x$ sandwich structure is constructed for a solid-state electrical gate to tune the conductivity of graphene (details of the fabrication can be seen in the Methods section). A 50-nm Al$_2$O$_3$ layer is used as a dielectric gate, which has a negligible effect on the light reflection. The back gate electrode is made of 10-nm thickness of a TiO$_x$ (sheet resistance ~2000 Ω) film. This thin and high-resistance film ensures the transparency for the THz light. The transparency of TiO$_x$ is verified by THz transmission spectrum and the loss of TiO$_x$ is also discussed in Supplementary Figure 3 and Supplementary Note 3. The properties of graphene are first characterized by atomic force microscopy (AFM) for surface morphology with measurement of the thickness and also by Raman spectrum (Supplementary Figure 4 and Supplementary Note 4). The tunability of graphene's

conductivity through the electrical gate is tested by the device's THz transmission spectra (Supplementary Figure 5 and Supplementary Note 5), as well as verified by a graphene field-effect transistor (G-FET) (Supplementary Figure 6 and Supplementary Note 6).

**Actively controlled Brewster angle.** In order to obtain the Brewster angle of the device, we measured the reflected THz pulse with an incident angle of 55–75°, as well as gate voltage from −12 to 12 V. Three typically reflected THz pulses in the time domain are shown in Fig. 2a–c. When the incident angle is 55°, the pulse keeps an asymmetric "M" shape from −12 to 12 V. The pulse amplitude decreases monotonically with the gate voltage. When the incident angle is 68°, the pulse shape changed from "M" to "W", indicating a significant phase change during this process. When the incident angle is 75°, the THz pulse keeps an asymmetric "W" shape from −12 to 12 V and the amplitude decreases monotonically. The pulse shape changing from "M" to "W" indicates that the phase of THz wave is changed by 180°. We plot the curve of the amplitude as a function of incident angle under different gate voltages in Fig. 2d (here the result of 0.8 THz is taken as an example). The results for other frequencies are shown in Supplementary Figure 7 and Supplementary Note 7. The "V" curves show similar trends to the theoretical results (Fig. 1b). To verify our physical model, the conductivity of graphene is extracted by fitting the experimental data to Eq. (1), as shown in Fig. 2d, and then the collected data are compared with the conductivity extracted from transmission spectra (Supplementary Figure S5). The fitted conductivity increases from 0.3 to 2.2 mS, and is consistent with the conductivity extracted from the device's THz transmission spectra (red curve of Fig. 2e), as well as that from G-FET (Supplementary Figure 6c). The fitting results for other frequencies are shown in Supplementary Figure 7f, which is also consistent with that extracted from transmission spectra. With the parameters extracted from fitting, the theoretical reflection amplitude as a function of incident angle for the frequency is achieved, as shown in Supplementary Figure 8 and Supplementary Note 8. Figure 2f is the Brewster angle as a function of gate voltage at different frequencies, and shows that the Brewster angle can be tuned from about 64.5 to 71.5°. The minute change in different frequencies is due to the weak dispersion relation of conductivity in this frequency region. The passive tuning of Brewster angle by metamaterials and metasurface has been reported in previous works[34,35]. However, these methods are hard to active tuning of Brewster angle. To change the Brewster angle, the unit structure or periodic parameter for the metamaterial or metasurface should be changed. Besides, Brewster angle of metamaterials/metasurface is always frequency-dependent, due to their resonance nature. As far as we know, this work is the first time to realize an active tunable Brewster angle and achieve broadband response.

**Intensity modulator.** From the result in Fig. 2c, we observe that a nearly zero reflection can be achieved at an incident angle of 65° when the gate voltage is 12 V. This indicates great potential to use our device as an intensity modulator with deep modulation depth at this incident angle. Figure 3a shows the evolution of the time-domain waveform reflected from the device as the voltage changed from −12 to 14 V. The waveforms maintain a similar shape, while the peak-to-peak value decreases from 1.37 to 0.067, indicating a nearly unchanged phase and a deeply modulated intensity. The maximum reflected pulse is achieved when the gate voltage is −12 V, and we define it as the baseline for calculating the modulation depth. The reflection amplitude can be achieved through Fourier transform (Supplementary Figure 9

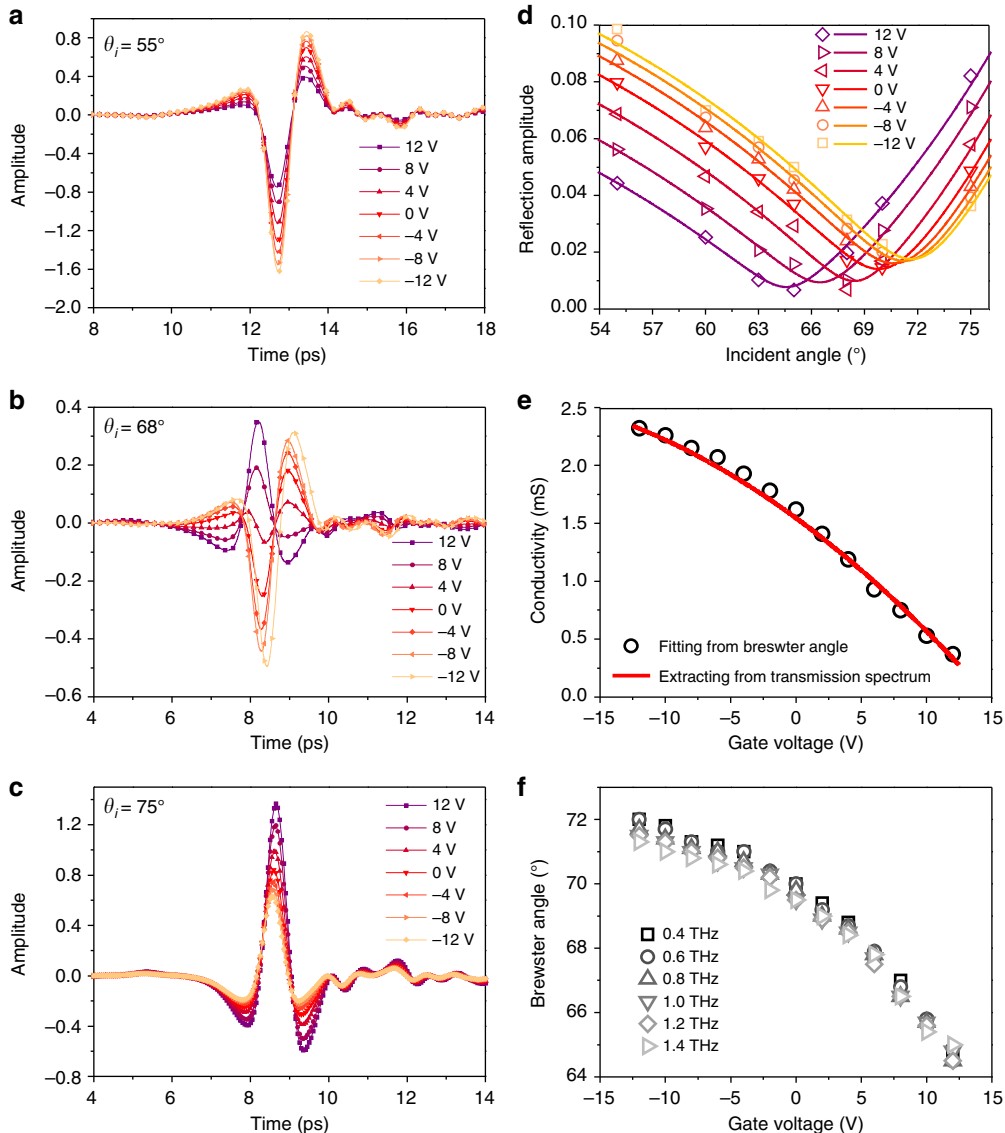

**Fig. 2** Controlling the Brewster angle by adjusting the conductivity of graphene. The reflection THz time-domain signal under different gate voltages with incident angles of (**a**) 55, (**b**) 68, and (**c**) 75°. **d** Reflection amplitude as a function of incident angle for the frequency of 0.8 THz. Open symbols represent the experimental amplitude and solid lines show the curve fitting according to Eq. (1). **e** The fitted conductivity for different gate voltages at the frequency of 0.8 THz (open circles) and the conductivity extracted from the transmission spectrum (red line). **f** The Brewster angle as a function of gate voltage at different frequencies

and Supplementary Note 9). The modulation depth of the peak to peak is up to 99.7%, as shown in the inset of Fig. 3a. By Fourier transformation of each signal and comparing them to the baseline, a modulation depth of higher than 99.3% is achieved over the frequency range of 0.5−1.6 THz and the maximum modulation depth is 99.9%, indicating the broadband and spectrally flat modulation. The high modulation depth is due to the nearly zero reflection of THz light near the Brewster angle. Experimental insertion loss of this intensity modulator is about ∼−12 dB (detailed calculation is shown in Supplementary Note 2 and Supplementary Figure 2), which is comparable with or better than that of the high-performance narrow bandwidth metadevices[38−40]. This experimental insertion loss is mainly due to the poor quality of commercial graphene and the fabrication process.

In order to further clarify this phenomenon, we simulated the electric field distribution at the frequency of 0.8 THz, when graphene's conductivity is 0.5 and 3.5 mS, corresponding to the

gate voltage of 14 and −12 V in the experiment. As shown in Fig. 3c, when the conductivity is low (0.5 mS), the incident angle is very close to the Brewster angle of the device, which results in a very weak reflection being observed. While graphene's conductivity is 3.5 mS, the Brewster angle is moved far away from the current incident angle, as shown in Fig. 2c. This results in a considerably strong reflection field, as illustrated in Fig. 3d. The fitted reflection amplitude as a function of gate voltage shows a high degree of matching with our theoretical results, as well as experimental results (as shown in Supplementary Figure 10 and Supplementary Note 10).

The performance of the device, both of the modulation depth and operation band, is much higher than that of previous results[6−13,23−30,41−43]. More importantly, owing to the solid-state-based modulations, our device provides a high modulation speed, as shown in Fig. 4a. The rising time of modulation is about 0.1 ms, as shown in Fig. 4b, corresponding to the modulation speed of about 10 kHz. When the modulation speed is up to

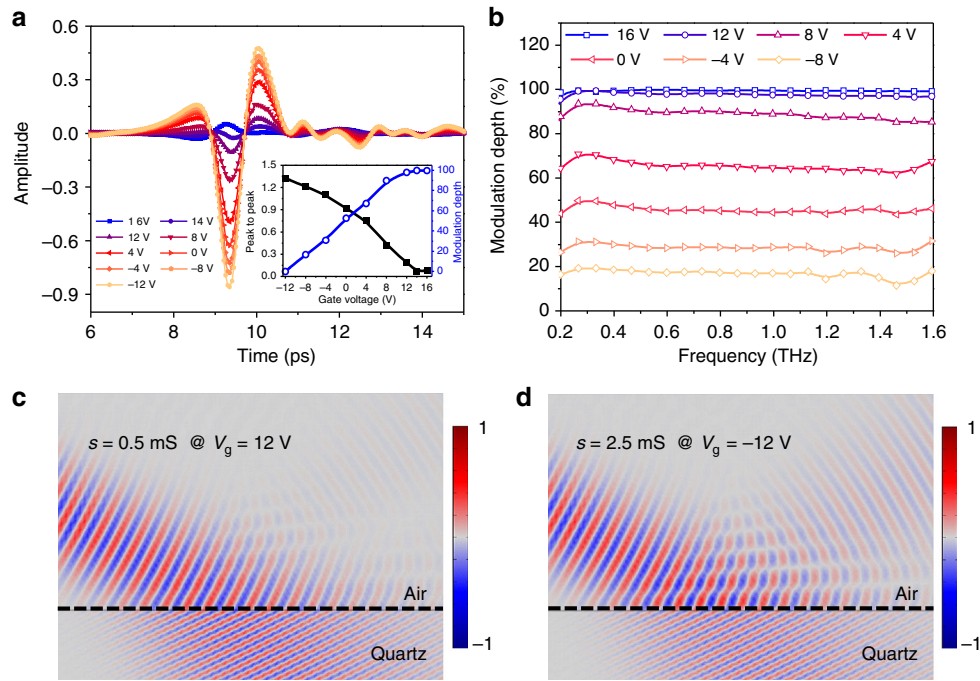

**Fig. 3** The device operates as a THz intensity modulator, when the incident angle is 65°. **a** Reflected THz signal in a time domain. **b** The modulation depth as a function of frequency. The simulated *E*-field with a conductivity of (**c**) 0.5 mS, and (**d**) 2.5 mS under the incident angle of 65°. The dashed line presents the graphene on quartz

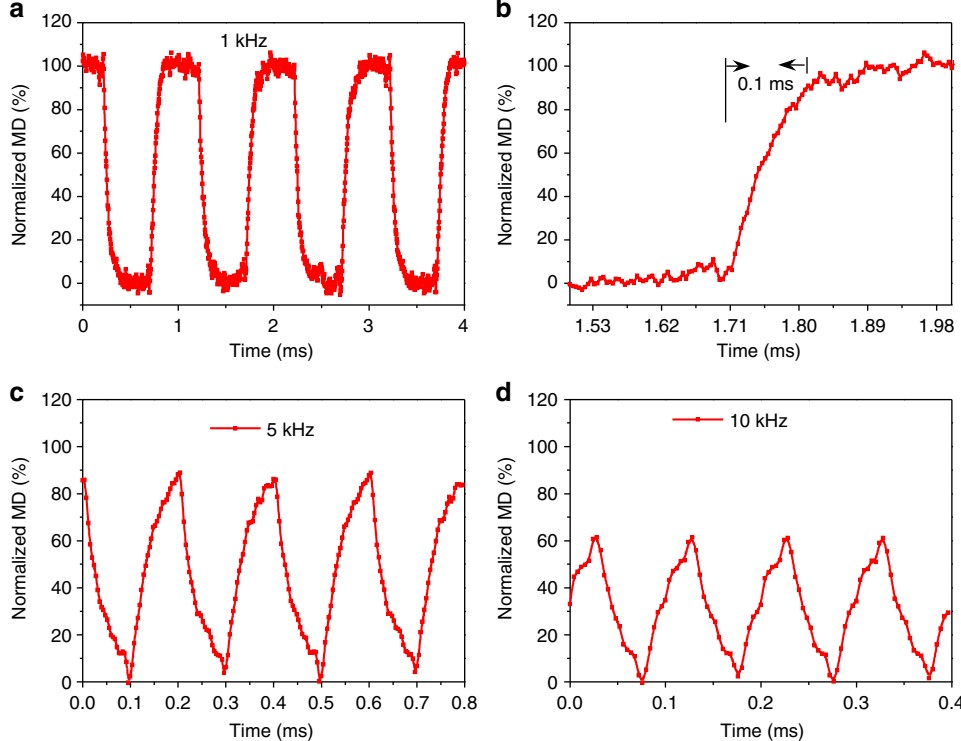

**Fig. 4** Modulation speed of the intensity modulator. **a** The modulated terahertz beam signal under the driving signal of 1-kHz square wave with ±10 V; (**b**) the zoom-in of the rising edge in (**a**); the normalized modulation depth under ±10-V square wave with (**c**) 5 kHz and (**d**) 10 kHz

5 and 10 kHz, the modulation depth decreases to 80% and 60%, respectively, as shown in Fig. 4c, d. Theoretically, the modulation speed of a solid-state dielectric-gating device is limited by the resistance–capacitance (RC) time constant. In our device, the effective series resistance (R) comes from the graphene and $TiO_x$.

Here, we take a resistance of 250 Ω for the graphene and 1000 Ω for the $TiO_x$, which are calculated by the half of averaged square resistance over the gate voltage range. A capacitance (C) of ~140 nF is calculated by adopting a 50-nm $Al_2O_3$ and an active device (graphene) area of 1.0 × 1.0 cm. Therefore, the calculated

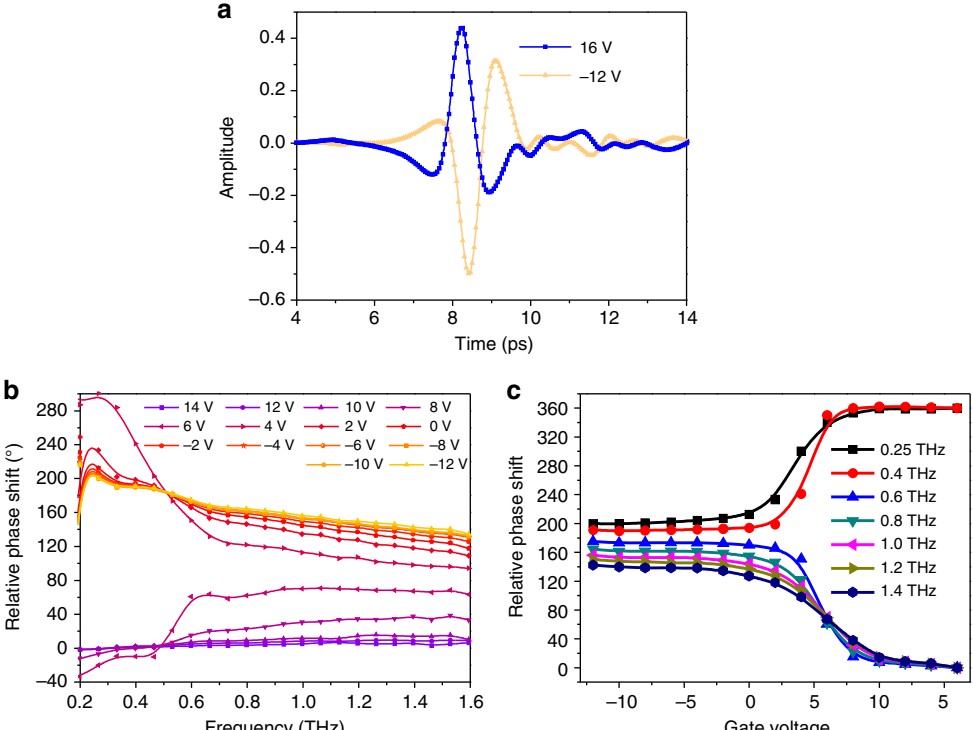

**Fig. 5** The device worked as a THz-phase modulator under an incident angle of 68°. **a** Reflected THz time-domain signal at the gate voltages of 16 and −12 V. **b** Relative phase shift referenced to the phase shift at 16 V from 0.2 to 1.6 THz. **c** Phase shift as a function of gate voltage at different frequencies

RC time constant is $1.75 \times 10^{-4}$ S, corresponding to a modulation speed of ~6 kHz, which agrees well with the experimental result. Higher modulation speed (shorter RC time constant) can be realized through replacing $TiO_x$ by a higher-conductivity material (e.g., graphene) and reducing the device size. For instance, by utilizing graphene to replace $TiO_x$ and reducing the device size to ~1.0 × 1.0 mm (about two times the wavelength of 0.8 THz), the modulation speed will be ~2.4 MHz, which is comparable with other solid-state THz modulators in previous reports[4,7,24,27].

**Phase modulator**. According to the physical model, a broadband phase modulation with a deep modulation range can be achieved by utilizing the phase-jump property across the Brewster angle. Here, we chose the incident angle of 68° to discuss this issue. The reflected THz time-domain signal changes from an asymmetry "W" shape at the gate voltages of 16 V to an asymmetry "M" shape at the gate voltages of −12 V, which indicates that there should be a large phase shift from $V_g = 16$ V to $V_g = -12$ V, as shown in Fig. 5a. The phase shift in the frequency domain is extracted from the time-domain pulse by Fourier transform and referenced to the phase at a gate voltage of 16 V, as shown in Fig. 5b. Over the broad frequency range from 0.5 to 1.6 THz, as the gate voltage decreased from −8 to 16 V, the relative phase shift decreases monotonically with a modulation range over 140°, showing that an ultra-broadband and deep phase modulation is achieved, which is highly consistent with our simulation results. We also noticed that there is a relatively larger modulation range at low frequencies (e.g., −180° at 0.4 THz) and a relatively smaller modulation range at high frequencies (e.g., 140° at 1.6 THz). This frequency dependency is expected to contribute to the frequency-dependent imaginary conductivity of graphene described by the Drude model (Supplementary Note 11 and Supplementary Figure 11).

## Discussion

In this work, THz modulation-based graphene-controlled Brewster angle device is experimentally demonstrated for the first time. The device is fabricated using a graphene/$Al_2O_3$/$TiO_x$ sandwich structure on a quartz substrate. The conductivity of the graphene can be electrically controlled to generate a variable Brewster angle. Based on this device, an ultra-broadband THz-intensity modulator with a modulation depth larger than 99.3% and an ultra-broadband-phase modulator with a tunability higher than 140° are demonstrated. Our work presents graphene-controlled Brewster angle device for ultra-broadband THz intensity and phase modulation at room temperature. The promising performance of our device also suggests that there exists a great potential to use graphene in deliberate architecture for active THz devices.

## Methods

**Device fabrication**. A Ti film of 8 nm was deposited on a quartz substrate (diameter of 2 cm) for the bottom-gate electrode. In order to avoid further reducing the conductivity of Ti, the Ti film was oxidized into $TiO_x$ through annealing at 150 °C. After oxidization, the conductivity of this film was about a reciprocal of sheet resistance of 2000 Ω, which was measured using a four-probe method. Then, a layer of $Al_2O_3$ (50 nm) was deposited by atomic layer deposition onto $TiO_x$ as a dielectric gate. Afterward, graphene (~1 × 1 cm) was transferred onto the as-prepared quartz/$TiO_x$/$Al_2O_3$ substrate. Subsequently, two electrodes (Cr/Au) were fabricated by thermal deposition. One was on graphene and the other one was on $TiO_x$ for the gate voltage. The device is schematically shown in Fig. 1f.

**Measurements**. A THz-TDS system from Menlo Systems (TERA-K15) was used for the terahertz reflection measurements. A reflection guide was used for manual adjustment of the reflection angle, ranging from a minimal reflection angle of 55 to 75°.

## Data availability

All data supporting the findings of this study are available within the article and its Supplementary Information. All other data are available from the corresponding author upon reasonable request.

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

## Acknowledgements

The work is in part supported by Research Grants Council of Hong Kong, particularly, via grant nos. 14207515, 14201415, 14204616, AoE/P-02/12, Hong Kong SAR Government, and CUHK Group Research Scheme, the Hong Kong Innovation and Technology Fund (project number ITS/371/16), and the Royal Society Wolfson Merit Award (E.P.-M.). J.-B.X. would like to thank the National Science Foundation of China for the support, particularly, via grant no. 61229401.

## Author contributions

Z.C. and J.-B.X. conceived the concept. Z.C. fabricated the devices, conducted the measurements, and analyzed the data. X.C. established the experimental setup and conducted the measurements, and contributed to the analysis. L.T., K.C., M.L., K.Y. and X.L. assisted in the fabrication and data analysis. R.I.S. developed and instructed the modulation speed measurement program. J.-B.X. and E.P.-M. supervised the work. All the authors discussed the work.

## Additional information

**Competing interests:** The authors declare no competing interests.

