## [Peer Review File · Nature Communications]

Reviewers' comments:

Reviewer #1 (Remarks to the Author):

The manuscript "Graphene-controlled Brewster Angle Device for Ultra-Broadband Terahertz Modulation" by Z. Chen et al. is focused on the development of highly tunable graphene THz modulators exploiting a quartz/graphene interface. The topic is certainly interesting, widely reported in the literature by many research groups in the past 5 years and intriguing from an application point of view.

Although the experiment is well conceived and the manuscript well written, the data require a deeper investigation / further analysis and some claims need to be justified and corroborated. I therefore think that the manuscript cannot be accepted in the present form, and a further evaluation is needed after addressing my comments below:

- 1) The core material is graphene. However, the manuscript does not provide any detail on the graphene employed. Authors should carefully detail: growth technique, dimension of the graphene surface, AFM tomography with measurement of the thickness, Raman spectra to evaluate the quality of graphene before and after Al₂O₃ deposition;
- 2) At line 62 the authors claim "the reported bandwidth in this work is mainly limited by the effective bandwidth of the THz system". This sounds a little bit strange since the bandwidth of the Menlo TeraK15 claimed in the manuscript is too large. I therefore want to see the measured reflection and transmission bandwidth of the employed spectrometer.
- 3) Line 79: why the authors employ in the model a mobility value significantly larger than that effectively measured in their graphene? It would be more correct to use the exact value extracted in their sample.
- 4) Modulation speed: the authors estimate the modulation speed from calculations. To claim that they are developing a solid-state device providing a high modulation speed, they should report here a measurement of the effective modulation speed of the device component.

Reviewer #2 (Remarks to the Author):

This work demonstrates the effect of tunable Brewster angle controlled by graphene. This is a well written paper describing an interesting approach towards realization of broadband THz modulation. Following are several minor comments:

- 1- Modulation Speed: besides theoretically estimating, it should be simple enough to experimentally measure it.
- 2- TiO_x, this film provides a sheet resistance of 2k Ohm, that is 0.5mS sheet conductivity. It is not clear to me that such a film will be transparent to THz light.
- 3- From an operations perspective please elaborate on the key differences of this work with your previous work: "Graphene Based Terahertz Light Modulator in Total Internal Reflection Geometry" Adv. Optical Mat. 2016. Also explain why this work should not be considered as incremental.

Reviewer #3 (Remarks to the Author):

Chen et al. reported that graphene can be used to control Brewster angle at THz frequency and subsequently use this as an THz modulator. Overall it is an interesting demonstration to the community but needs to address the following concerns. From the point of view of applications, I think this idea is hard to be adopted in a practical device because it uses reflection geometry (in comparison with transmission geometry).

- 1) I can imagine that some other methods (metamaterials etc) have been used do the active/passive tuning of Brewster angle. In comparison with these previously proposed methods,

what are the advantages of using graphene as the active medium as shown in this work? I would like to suggest the authors to make this clear to the readers.

2) If using such a system as an intensity modulator, the 'loss' (insertion loss + transmitted intensity) is huge because of the poor reflectivity of graphene (after all it is only one monolayer of carbon atoms). To enhance reflectivity, the more conductive medium is needed at the interface, but this will then lead to more insertion loss of energy. There should be existing a maximum reflection percentage of such structures ($1 - \text{minimum}(\text{insertion loss} + \text{transmission})$), so what is this value both experimentally and theoretically? Would it be more beneficial to use more layers of graphene?

3) What's the current device size demonstrated in this work?

4) Authors suggest that a graphene layer can be used to replace the bottom TiO_x to increase the device speed. However, this double graphene layer structure may cause extra reflection from the bottom graphene layer? Will the distance between two graphene layers eventually affect the reflection efficiency? Please briefly comment on this in order to evaluate this double graphene layer structure.

5) The manuscript is well written but the figures in the manuscript are poorly displayed. For example, the figure 1 is hard to read, specially the figure legends etc (fonts are too small). Also, please keep consistent with the color scheme throughout the manuscript. For example, the color used for one certain gate voltage reflection curves vary from figure to figure, which makes it hard to read. Please keep consistent.

Reviewers' comments:

Reviewer #1 (Remarks to the Author):

The manuscript "Graphene-controlled Brewster Angle Device for Ultra-Broadband Terahertz Modulation" by Z. Chen et al. is focused on the development of highly tunable graphene THz modulators exploiting a quartz/graphene interface. The topic is certainly interesting, widely reported in the literature by many research groups in the past 5 years and intriguing from an application point of view.

Although the experiment is well conceived and the manuscript well written, the data require a deeper investigation / further analysis and some claims needs to be justified and corroborated. I therefore think that the manuscript cannot be accepted in the present form, and a further evaluation is needed after addressing my comments below:

1) The core material is graphene. However, the manuscript does not provide any detail on the graphene employed. Authors should carefully detail: growth technique, dimension of the graphene surface, AFM tomography with measurement of the thickness, Raman spectra to evaluate the quality of graphene before and after Al₂O₃ deposition;

Response: thank the suggestions. The CVD graphene is a commercial product, which is purchased from the company named 'SixCarbon Technology (Shenzhen)'. The Raman spectra are shown in figure R1 (a). After transferring the graphene on to the Si/SiO₂ wafer, there were three characteristic peaks: D, G and 2D peaks. When the graphene was transferred to our prepared substrate, quartz/TiO_x/Al₂O₃, we can observe not only graphene's three characteristic peaks, but also some other peaks contributed to the substrate (the black line). For graphene on the substrate of quartz/TiO_x/Al₂O₃, the intensity of D peak is very low and the intensity of 2D is two times over that of G. These indicate graphene is single layer with low defect. The tomography of graphene (figure R1(b)) shows that the surface of graphene is very flat, except some folds due the stress during the transfer process. To characterize the thickness of graphene, we measure tomography of graphene at the edge of the film, where graphene fragment can be found. The thickness of graphene is about 1.2nm, which is thicker than the intrinsic thickness. This is because of some fine PMMA residue and contaminants on the graphene surface increase the thickness. The characterization of Raman spectrum, and AFM is added in supporting information S4.

Figure R1. (a) Raman spectrum of quartz/TiO_x/Al₂O₃ and graphene on quartz/TiO_x/Al₂O₃ substrate and Si/SiO₂ wafer. (b) and (c) tomography of graphene.

2) At line 62 the authors claim “the reported bandwidth in this work is mainly limited by the effective bandwidth of the THz system”. This sound a little bit strange since the bandwidth of the Menlo TeraK15 claimed in the manuscript is too large. I therefore want to see the measured reflection and transmission bandwidth of the employed spectrometer.

Response: Thank you for the comments.

The bandwidth of the Menlo TeraK15 can be up to maximum 4.5 THz. However, the spectrum has a maximum signal-to-noise ratio (SNR) at around 0.5 THz and it rapidly drops with the increasing of frequency after that. The actual effective spectrum highly depends on the optical setup and configuration. For example, using metallic parabolic mirror provides much better SNR than using convex TPX lenses (which are what we used). It also depends on the laser power and voltage applied. The spectrum of our K15 system in the system test report is shown in Fig R2. The 40 dB power loss point is found at 1.9 THz. Spectrum higher than 1.9 THz has a SNR lower than 15 dB, which can be easily affected by the insertion loss of the polarizers and the device.

The actual transmission and reflection (68° incident) spectrum of our experiments are shown in figure R3 (a) and (c), while the relative reflection shown in Fig. R3 (b) and (d). The spectrum distribution matches well with Fig. R2. An obvious improvements on the modulation ability can be found on the reflection spectrum compared to the transmission results. Specially, the SNR is further reduced. Therefore 1.6 THz bandwidth is a reasonable range for evaluating the device performance. To make it clear, we have added the corresponding reflection amplitude in supporting information S8.

Frequency Domain Spectra

Signal to noise ratio		
Signal to Noise Ratio	> 70 dB	Passed
THz Bandwidth	> 3.5 THz	Passed

Place, Date: Menlo Systems, August, 1st 2013
Technician: Rafal Wilk

Fig. R2 Menlo K15 spectrum of the system applied in the system test report

Figure R3 (a) transmission amplitude extracted from THz time dominate pulse through Fourier transform. (b) relative transmittance. (c) reflection amplitude extracted from THz time dominate pulse through Fourier transform. (d) relative transmittance.

3) Line 79: why the authors employ in the model a mobility value significantly larger than that effectively measured in their graphene? It would be more correct to use the exact value extracted in their sample.

Response: The logic of this manuscript is: firstly we propose a physical phenomenon (tunable Brewster angle) and its potential application, and then we design devices to verify the physical model, at last we use the device for intensity and phase modulation. We had not fabricated the samples when we proposed the model and experimental values were not available. Therefore we used theoretical values for evaluating the device performance.

The exact value of graphene's parameter is achieved when fitting the measurement results to the Brewster reflection model, also can be extracted from the transmission spectrum, as well as field effect transistor (FET). The mobility is about $1.3 \times 10^3 \text{ cm}^2/(\text{V} \cdot \text{s})$ and the conductivity ranging from about 0.4mS to 2.4mS. The theoretical results are shown as figure R4, when using the exact value extracted from our sample. We add this results in supporting information S11.

Figure R4. Theoretical result of tunable Brewster angle when $\mu=1.3 \times 10^3 \text{ cm}^2/(\text{V} \cdot \text{s})$, the exact mobility of graphene sample in the experiment.

4) *Modulation speed: the authors estimate the modulation speed from calculations. To claim that they are developing a solid-state device providing a high modulation speed, they should report here a measurement of the effective modulation speed of the devices component.*

Response: thank you for the comments. We have added the measurement of modulation speed as figure 4 in manuscript, as well as shown in figure R5 of this response letters. The modulation speed of our device is about 10kHz (modulation depth decrease to 50%), which agrees well with the resistance-capacitance (RC) time constant (1.75×10^{-4} s). Higher modulation speed (shorter RC time constant) can be realized through reducing the device size and reducing the contact resistance, as well as increasing the conductivity of graphene and TiOx.

The modulation speed is added in the manuscript figure 4, and relative discussion is added and highlighted in page 9.

Figure R5. (a) the modulated terahertz beam signal under the driving signal of 1kHz square-wave with +/- 10V; (b) the zoom-in of rising edge in (a); (c) and (d) driving signal with 5kHz and 10kHz respectively.

Reviewer #2 (Remarks to the Author):

This work demonstrates the effect of tunable Brewster angle controlled by graphene. This is a well written paper describing an interesting approach towards realization of broadband THz modulation. Following are several minor comments:

1- Modulation Speed: besides theoretically estimating, it should be simple enough to experimentally measure it.

Response: thank you for the comments. We have added the measurement of modulation speed as figure 4 in manuscript, as well as shown in figure R5 of this response letters. The modulation speed of our device is about 10kHz, which agrees well with the resistance-capacitance (RC) time constant. Higher modulation speed (shorter RC time constant) can be realized through reducing the device size and reducing the contact resistance, as well as increasing the conductivity of graphene and TiOx.

The modulation speed is added in the manuscript figure 4, and relative discussion is added and highlighted in page 9.

Figure R5. (a) the modulated terahertz beam signal under the driving signal of 1kHz square-wave with +/- 10V; (b) the zoom-in of rising edge in (a); (c) and (d) driving signal with 5kHz and 10kHz respectively.

2- TiOx, this film provides a sheet resistance of 2k Ohm, that is 0.5mS sheet conductivity. It is not clear to me that such a film will be transparent to THz light.

Response: The transparency of TiOx can be characterized by transmission measurement and also can be estimated by Fresnel equation. The THz signal and corresponding transmission spectrum are shown in figure R6 (a) and (b). The transmission is about 0.9 over the frequency of 0.2THz to 1.6THz. According to Fresnel equation, the transmission of a free standing ultra-thin conducting film in air is $t = \frac{2}{2+Z_0\sigma}$, where $\sigma = 0.5\text{mS}$ is the conductivity of the TiOx (measured by four probes) and $Z_0 = 377$ is the impedance of air. So we can estimate the transmittance of the TiOx is about 0.91, which is consistent well with the experimental result. The discussion about the transparency of the TiOx in THz is added as supporting information S3.

Figure R6 Transmission of TiOx on Quartz (a) time domain spectrum (b) transmission spectrum in frequency domain.

3- From an operations perspective please elaborate on the key differences of this work with your previous work: "Graphene Based Terahertz Light Modulator in Total Internal Reflection Geometry" Adv. Optical Mat. 2016. Also explain why this work should not be considered as incremental.

Response: Thank you for the comments.

There are three main difference of this work comparing with our previous works:

(a) The working mechanism is totally different. In our previous work, the device is based on total internal reflection; while in this work, the THz modulator is based on graphene-controlled Brewster angle, which have never been discussed before. Specially, in this work we demonstrated that the Brewster angle can be electronically controlled by graphene.

(b) The device structure is simplified. In previous work, the device must be coupled with a prism, which has more critical requirements on the optical alignment. More importantly, the device based on internal total reflection requires highly tunable conductivity (about 0.2mS to 8mS) of graphene to get high modulation depth. To satisfy this condition, ion gel gate is introduced into the device, which dramatically limits the modulation speed. In this work, because of the different working principle, tunable conductivity (about 0.2mS to 2.5mS) of graphene, which is easily to be achieved by normal solid gating, can realize a modulation depth up to 99.7%. Besides, the device can work directly without prism.

(c) The performance is drastically improved. In previous work, the modulation depth is about 90% with ion gel gate, while it is 99.7% with solid state gate in this work due to their different working mechanism. Besides, it is well known that the modulation speed of the device with ion gel gate is much lower than the device with solid state gate. It has been reports that the modulation speed of a dozens-micron size field effect transistor with vertical ion gel gate is around 1kHz (Nature Materials,7, 900–906 (2008)). So it can be estimated that the speed of a millimeters size device with ion gel side gate should be down to several Hz. In this work, the modulation speed is upto 10kHz and can be further improved through device optimization.

In a word, from the operation perspective the Brewster device shows more excellent performance (modulation depth and speed) and simpler device structure than that based on total internal reflection geometry.

Reviewer #3 (Remarks to the Author):

Chen et al. reported that graphene can be used to control Brewster angle at THz frequency and subsequently use this as an THz modulator. Overall it is an interesting demonstration to the community but needs to address the following concerns. From the point view of applications, I think this idea is hard to be adopted in a practical device because it uses reflection geometry (in comparison with transmission geometry).

Response: We thank the reviewer for the valuable comments. We agree that reflection geometry is harder than transmission geometry to be adopted in a practical device. However, compared to most other reflection-based devices which usually have a polarization-sensitive metamaterial surface, or a requirement on a prism, our device is much simpler in assembling into a system. The alignment of the device is similar to aligning a mirror. Giving that the performance of modulation (both intensity and phase) is much better than that of the transmission geometry, we believe it has wide application potentials. Besides, similar to the Brewster device in visible light, active Brewster device in THz region makes it possible for special THz optical components, such as active attenuator, polarizing filter, et al.

1) *I can imagine that some other methods (metamaterials etc) have been used do the active/passive tuning of Brewster angle. In comparison with these preciously proposed methods, what are the advantages of using graphene as the active medium as shown in this work? I would like to suggest the authors to make this clear to the readers.*

Response: thank you for your comment. The research of Brewster angle on metamaterials and metasurface have been reports in previous works. Fundamentally, Brewster angle is depended on the permittivity, permeability, and surface conductivity. The following are some examples of Brewster angle on metamaterials and metasurface:

Brewster's effect for transverse-electric waves in GHz frequency by constructing metamaterials with non-unity permeability; this phenomenon also been found in visible light region in metal-dielectric metamaterial [*Phys. Rev. B* 73, 193104(2006)]. Andrea Alu *et al.* propose plasmonic Brewster Angle which can be controlled by parameters of optical grating (such as width, filled factor) [*Phys. Rev. Lett.* 106, 123902(2011)]. Ramón Paniagua-Domínguez et al proposed that Brewster's effect potentially can be design for any angle, wavelength, and polarization of choice by all-dielectric metasurface [*Nature Communications*, 7, 10362 (2016)].

However, these methods are hard to realized active tuning of Brewster angle. To change the Brewster angle, new unit structure or periodic parameter is required for the metamaterial or metasurface. Another disadvantage is that Brewster angel based on metamaterials/metasurface is always frequency-dependent, due to their resonance nature. Comparing with the previous works, using graphene as the active conducting surface can realized electrical tuning of Brewster angle and achieved broadband response.

We add the following section in discussion of **tunable Brewster angle** in page 6:

'The passive tuning of Brewster angle by metamaterials and metasurface have been reports in previous works³⁸⁻⁴⁰. However, these methods are hard to realized active tuning of Brewster angle. To change the Brewster angle, new unit structure or periodic parameter is required for the metamaterial or metasurface. Besides, Brewster angel of metamaterials/metasurface is always frequency-dependent, due to their resonance nature. As far as we know, this work is the first time to realize active tunable Brewster angle and achieve broadband response.'

2) If using such a system as an intensity modulator, the 'loss' (insertion loss + transmitted intensity) is huge because of the poor reflectivity of graphene (after all it is only one monolayer of carbon atoms). To enhance reflectivity, the more conductive medium is needed at the interface, but this will then lead to more insertion loss of energy. There should be existing a maximum reflection percentage of such structures (1-minimum(insertion loss+transmission)), so what is this value both experimentally and theoretically? Would it be more beneficial to use more layers of graphene?

Response: Thank you for the comments. We agree that the poor reflectivity of graphene lead to large 'loss' (insertion loss + transmitted intensity) and the more conductive medium are needed to enhance reflectivity. Exactly, better tunability on the conductivity of the medium is needed. However, the insertion loss does not increase monotonically as the increase of conductivity. Insertion loss due to the resistance of a conducting layer can be expressed by $L_{insertion} \propto \sigma \cdot E^2$. As the conductivity increases, the conducting layer surface reflects more light which allows less E-field to penetrate into the conducting layer. The E term in the above equation reduces with the increasing of the conductivity, resulting in a conductivity point of 11mS where the maximum insertion loss can be found, as shown in Fig. R7 (a). The overall reflection of the device calculated by Eq. 1 in the manuscript represents the ultimate reflection which has taken the insertion loss and transmission loss into account (1 - insertion loss - transmission loss, i.e. the 'reflection percentage' mentioned in the question). As can be seen from the blue curve in Fig. R7 (a), the reflection monotonically increases with the conductivity. Considering the Drude model of graphene, we can achieve the insertion loss and reflection as a function of the graphene Fermi level (Fig. R7 (b)), which shows similar results. Therefore, the device always benefits from larger conductivity tunability.

Increasing the layer number of graphene is a very good idea to improve the performance of the device. In figure R7(c), we show the calculation results for using different graphene layers. More calculation results for improving the reflectance of the device is shown in supporting information S2.

figure R7 (a) insertion loss and reflection amplitude as a function of the conductivity of conducting layer on quartz. (b) Insertion loss and reflection amplitude as a function of Fermi level of graphene on quartz surface. The conductivity of graphene is described by Drude model. (d) Reflection amplitude with different graphene layer placing on quartz.

3) *What's the current device size demonstrated in this work?*

Response: thank you for the comment. The device is fabricated on round quartz with diameter of 1.5 cm. an active device (graphene) area of 1.0 cm ×1.0 cm. This information is added in the method section.

4) *Authors suggest that a graphene layer can be used to replace the bottom TiOx to increase the device speed. However, this double graphene layer structure may cause extra reflection from the bottom graphene layer? Will the distance between two graphene layers eventually affect the reflection efficiency? Please briefly comment on this in order to evaluate this double graphene layer structure.*

Response: thank you for the comment. In double graphene layer structure, the bottom graphene layer dose cause extra reflection. However, the reflection from bottom layer graphene is benefit for the performance of device. The bottom layer graphene not only works as an electrode, but an tunable conducting layer, which tuned by the top graphene; similarly, the top layer graphene not only work as tunable conducting layer, but also an gate electrode for bottom layer graphene. Therefore, in double graphene layer structure, both of the graphene layers are tuned by gate voltage. Since hole and electron mobilities and their density of states are almost the same due to the graphene symmetric band structure, each graphene layer can contribute equally to terahertz modulation. Therefore, the double graphene layers structure can help overcome the potentially limited modulation in a single graphene layer. The energy-band diagram of double layer graphene is shown in figure R7. This enhancement effect for graphene's conductivity can be discussed in [*Nature Communications* 3, 780 (2012)].

Another key parameter is the distance between two graphene layers. The distance must be far smaller than the wavelength of THz wave, so that the interference effect on two graphene layer can be ignored.

Figure R7 Energy-band diagram of double layer graphene sandwiching a dielectric layer.

5) *The manuscript is well written but the figures in the manuscript are poorly displayed. For example, the figure 1 is hard to read, specially the figure legends etc (fonts are too small). Also, please keep consistent with the color scheme throughout the manuscript. For example, the color used for one certain gate voltage reflection curves vary from figure to figure, which makes it hard to read. Please keep consistent.*

Response: Thank you for the suggestions. We have modified all the figures.

Reviewers' comments:

Reviewer #1 (Remarks to the Author):

The authors addressed properly my comments and remarks.
The manuscript can now be accepted on Nature Communications

Reviewer #2 (Remarks to the Author):

The authors have addressed my previous comments in their revised manuscript. However, there are a few minor points that are still not clear thus benefit from additional discussion / modifications:

1- When computing for Modulation depth, the authors base their definition on power, however when defining transmission in Fig. R6 in accordance to the formula $t = 2 / (2 + Z_0 \cdot \sigma)$ they use electric field. Unless there is any particular reason for doing this, it seems to me, for the sake of consistency, better to always define and compute reflection and transmission in terms of power.

2- From the perspective of 1, the TiOx layer will introduce $100 \cdot (1 - 0.9^2) \sim 20\%$ of power to be reflected / absorbed. This non-transparency and related effects should be accounted for when computing and discussing the loss in the device.

Reviewer #3 (Remarks to the Author):

The authors revised the manuscript based on reviewers' suggestions. I recommend its its publication as it is.

Reviewers' comments:

Reviewer #2 (Remarks to the Author):

The authors have addressed my previous comments in their revised manuscript. However, there are a few minor points that are still not clear thus benefit from additional discussion / modifications:

1- When computing for Modulation depth, the authors base their definition on power, however when defining transmission in Fig. R6 in accordance to the formula $t = 2 / (2 + Z_0 \sigma)$ they use electric field. Unless there is any particular reason for doing this, it seems to me, for the sake of consistency, better to always define and compute reflection and transmission in terms of power.

Response: Thank you for the comments. We agree that it is more reasonable to define and compute reflection and transmission in terms of power. The transmission of TiOx has been revised accordance to the formula

$$T = |t|^2 = \left| \frac{2}{2 + Z_0 \sigma} \right|^2$$

and the corresponding results are shown in supporting information S3.

2- From the perspective of 1, the TiOx layer will introduce $100 * (1 - 0.9^2) \sim 20\%$ of power to be reflected / absorbed. This non-transparency and related effects should be accounted for when computing and discussing the loss in the device.

Figure R . (a) Calculation results of reflection for Brewster device with and without TiOx (b) reflection in term of power (c) reflection in dB scale

Response: Thank you for the comments. We agree that the TiOx layer will introduce 20% of power (about -1dB) to be absorbed/reflected at the 0° normal incident angle. However, our device is working at Brewster angle, so the loss is different. To clear the influence of TiOx on our device, we calculate the reflectance of Brewster device with and without TiOx. In the calculation, the conductivity of TiOx is set to be 0.5mS; the conductivity of graphene is set to be from 0mS to (3.7+7.5i) mS, which corresponds to the Fermi energy from 0eV to 0.4eV. Figure R(a) is the reflection amplitude for the device with and without TiOx when the graphene conductivity is at 0 mS and (3.7 + 7.5i) mS. When the conductivity of graphene is zero, the Brewster angle (where the reflection amplitude is minimum) for the device with TiOx is shifted from 63° to 65° due to the conductivity of TiOx. This indicates that when using this device with TiOx for an intensity modulator, the incident angle should be 65°.

When the conductivity of graphene increases to (3.7 + 7.5i) mS, the Brewster angle for the devices are almost the same and the minimum reflection amplitude for the device with TiOx is little smaller than that without TiOx. More importantly, at the incident angle of 65°, the reflection amplitude for the device with and without TiOx is almost the same. In the term of power $|r|^2$, they are also almost the same (Figure R(b)). This means that the TiOx of 0.5mS do not influence the modulation depth. Figure R(c) are the reflection in dB scale, where the

reflection at $\sigma_g = (3.7 + 7.5i)$ mS is corresponding to the insertion loss of the device. We can see that the device with TiOx does not suffer significantly increased loss (or very little loss) compared to the device without TiOx. The loss due to a conducting layer (TiOx and graphene, here) can be expressed by $L_{\text{insertion}} \propto \sigma \cdot E^2$, where σ is the total conductivity. On the other hand, when the conductivity increases, the conducting layer surface reflects more THz light and less E-field penetrates into the conducting layer, which decreases the E term in the above equation. As a result, introducing the TiOx layer does not cause palpable loss in the device.

In a short, the TiOx layer dose introduce 20% of power (about -1dB) to be absorbed when THz wave passes through the TiOx layer. However, in our Brewster device, the TiOx layer does not introduce palpable influence on both modulation depth and loss.

The discussions of the loss in TiOx (shown in the following) are modified/added in supporting information S2.

Figure S3 Transmission of TiOx on Quartz (a) time domain spectrum (b) transmission spectrum in frequency domain. (c) Calculation results of reflection for Brewster device with and without TiOx (d) reflection in term of power (e) reflection in dB scale

The THz signals for quartz and quartz/TiOx are shown in figure S3 (a), which show the same pulse shape. The corresponding transmission is about 0.8 over the frequency of 0.2THz to 1.6THz, figure S3 (b). According to Fresnel equation, the transmission of a free standing ultra-thin conducting film in air is $T = |t|^2 = \left| \frac{2}{2+Z_0\sigma} \right|^2$, where $\sigma=0.5\text{mS}$ is the conductivity of TiOx and $Z_0 = 377$ is the impedance of air. So we can estimate the transmission of TiOx is about 0.8, which corroborates well with the experimental result.

To clear the influence of TiOx on our device, we calculate the reflectance of Brewster device with and without TiOx. In the calculation, the conductivity of TiOx is set to be 0.5mS; the conductivity of graphene is set to be from 0 mS to $(3.7+7.5i)$ mS, which corresponds to the Fermi energy from 0eV to 0.4eV. Figure R(a) is the reflection amplitude for the device with and without TiOx when the graphene conductivity is at 0 mS and $(3.7 + 7.5i)$ mS. When the conductivity of graphene is zero, the Brewster angle (where the reflection amplitude is minimum) for the device with TiOx is shifted from 63° to 65° due to the conductivity of TiOx. This indicates that when using this device with TiOx for an intensity modulator, the incident angle should be 65° .

When the conductivity of graphene increases to $(3.7 + 7.5i)$ mS, the Brewster angle for the two devices are almost the same and the minimum reflection amplitude for the device with TiOx is little smaller than that without TiOx. More importantly, at the incident angle of 65° , the reflection amplitude for the device with and without TiOx is almost the same. In the term of power $|r|^2$, they are also almost the same (Figure R(b)). This means that the TiOx of 0.5mS do not influence the modulation depth. Figure R(c) are the reflection in dB scale, where the reflection for $\sigma_g = (3.7 + 7.5i)$ mS is corresponding to the insertion loss of the device. We can see that the device with TiOx does not suffer significantly increased loss (or very little loss) compared to the device without TiOx. The loss due to a conducting layer (TiOx and graphene, here) is proportional to the conductivity. $L_{\text{insertion}} \propto \sigma \cdot E^2$, where σ is the total conductivity. On the other hand, when the conductivity increases, the conducting layer surface reflects more THz light and less E-field penetrates into the conducting layer, which decreases the E term in the above equation. As a result, introducing the TiOx layer does not cause palpable loss in the device.

In short, the TiOx layer does introduce 20% of power (about -1dB) to be absorbed/reflected when THz wave passes through the TiOx layer. However, in our Brewster device, the TiOx layer does not introduce palpable influence on both modulation depth and loss.

REVIEWERS' COMMENTS:

Reviewer #2 (Remarks to the Author):

The authors have addressed all my previous comments in their revised manuscript. I recommend the manuscript for publication.